# A New 3D Image Block Ranking Method Using Axial, Coronal, and Sagittal Image Patch Rankings for Explainable Medical Imaging

## Abstract

Although a 3D Convolutional Neural Network (CNN) has been applied to explainable medical imaging in recent years, understanding the relationships among input 2D image patches, input 3D image blocks, extracted feature maps, top-ranked features, heatmaps, and final diagnosis remains a significant challenge. To help address this important challenge, firstly, we create a new 2D Grad-CAM-based method using feature selection to produce explainable 2D heatmaps with a small number of highlighted image patches corresponding to top-ranked features. Secondly, we design a new 2D image patch ranking algorithm that leverages the newly defined feature matrices and relevant statistical data from numerous heatmaps to reliably rank axial patches, coronal patches, and sagittal patches. Thirdly, we create a novel 3D image block ranking algorithm to generate a "Block Ranking Map (BRM)" by using the axial patch ranking scores, coronal patch ranking scores, and sagittal patch ranking scores. Lastly, we develop a hybrid 3D image block ranking algorithm to generate a reliable hybrid BRM by using different block ranking scores generated by the 3D image block ranking algorithm using different top feature sets. The associations between brain areas and a brain disease are reliably generated by using hybrid information from ChatGPT and relevant publications. The simulation results using two different 3D data sets indicate that the novel hybrid 3D image block ranking algorithm can identify top-ranked blocks associated with important brain areas related to AD diagnosis and autism diagnosis. A doctor may conveniently use the hybrid BRM with axial, coronal, and sagittal views to better understand the relationship between the top-ranked blocks and medical diagnosis, and then can efficiently and effectively make a rational and explainable medical diagnosis.

## 1 Introduction

In recent years, explainable deep learning techniques have been used in 3D medical imaging applications such as explainable 3D brain imaging. The explainable 3D CNN model with the gradient-weighted class activation mapping is made to predict prognosis using whole-body diffusion-weighted MRI data (Morita et al., 2024). The efficient methods are developed to generate visual explanations from 3D CNNs for Alzheimer's disease (AD) classification (Yang et al., 2018). In addition to the above mentioned methods using 3D information, another approach to increasing explainability of a 3D CNN is using relevant 2D information to interpret decisions of the 3D CNN. For instance, A 2D transformer-based medical image model with different transformer attention encoders is made to diagnose AD in 3D MRI images (Wang et al., 2024). However, two remaining significant challenges include (1) the fundamental research problem that is how to rationally interpret the relationship among 3D image blocks, extracted feature maps, important features, and final decisions of a 3D deep learning model, and (2) the fundamental application problem that is how to let a doctor conveniently understand such a multi-domain relationship and then efficiently make an explainable and correct medical diagnosis.

Currently, increasing explainability of a 3D deep learning model is important for high-quality medical imaging applications. For example, it is helpful and convenient for a medical doctor to easily see top-ranked visualized 3D brain image blocks that are closely associated with a brain disease such as

AD, and then make a correct and explainable brain disease diagnosis. In addition, it is useful to identify the most important blocks related to the brain disease in large 3D brain regions. For instance, it is important to identify top-ranked 3D blocks within the large Hippocampus. Thus, a fundamental problem is how to effectively select top-ranked 3D image blocks, and then better understand the relationship between the top image blocks and decisions of a 3D CNN.

Recently, various methods using 2D images have been developed to improve explainability of deep learning models. Relevant intelligent 2D techniques are developed to explain the decisions of a 2D CNN (Zhou et al., 2016a; Selvaraju et al., 2017; Zhang et al., 2018; Schöttl, 2022; Wang et al., 2022). For example, the Grad-CAM applies gradients to generate heatmaps for visual explanations of a deep neural network (Selvaraju et al., 2017). An efficient patch-based deep learning network with explainable 2D patch localization and selection is created for AD diagnosis (Zhang et al., 2023a). The feature selection (FS) is useful in not only improving the model performance but also in interpreting a deep neural network. For example, MediMLP with FS using Grad-CAM was developed for lung cancer postoperative complication prediction (He et al., 2019). Based on current explainable machine learning methods, we propose a novel method using axial, coronal, and sagittal 2D images extracted from 3D images to rank 3D image blocks.

For the two remaining significant challenges, we have new works described as follows. In section 2, new feature matrices with the top-ranked features' properties are defined, and then the new FS-Grad-CAM method is developed to generate explainable heatmaps with a smaller number of highlighted areas associated with top-ranked features. Also, a new 2D image patch ranking algorithm using both top-ranked features' properties and relevant statistical information in a large number of heatmaps is developed to reliably rank image patches. In section 3, a novel hybrid 3D image block ranking algorithm using the axial, coronal, and sagittal patch ranking scores is created to robustly rank 3D image blocks in order to allow a user to more easily understand the relationship between the top-ranked blocks and a decision. In section 4, simulation results are analyzed. In section 5, conclusions are given. In section 6, future works are discussed.

## 2 A New 2D Image Patch Ranking Algorithm

The last Maxpooling layer of a CNN generates $n$ $H \times W$ feature maps $F^l$ with the shape $H \times W \times n$ for $l = 0, 1, \ldots, n - 1$. A $\bar{H} \times \bar{W}$ input image has $P$ $(\bar{H}/H) \times (\bar{W}/W)$ patches for $P = HW$ (assuming $\bar{H}$ is divisible by $H$ and $\bar{W}$ is divisible by $W$). The feature maps $F^l$ have features $f_{ij}^l$ that are associated with a patch at $(i, j)$ for $i = 0, 1, \ldots, H - 1$, $j = 0, 1, \ldots, W - 1$, and for $l = 0, 1, \ldots, n - 1$. The $n$ feature maps are converted to $m$ flattened features for $m = n \times H \times W$. The $m$ features have $m$ feature index numbers $(0, 1, \ldots, m - 1)$.

An extracted feature map has $m$ flatten features that are associated with the relevant $m$ input image patches, and the $m$ flatten features are used by a classifier as inputs to make decisions. Since all $m$ flatten features in all extracted feature maps are used by the classifier, all associated patches have the same number of associated features. Because all patches are equally important for final decisions, the patches associated with all features cannot be ranked based on the number of associated features.

We developed a new CNN with the FS layer that applies a FS method to generate a small number of top-ranked features that are associated with a small number of patches. Other patches with 0 associated features are eliminated since they are not useful for decision-making, so they are not used for further patch ranking. Importantly, the FS method not only selects top-ranked features, but also identifies important associated patches. Thus, the identified patches associated with the top-ranked features can be ranked based on the number of associated features.

In addition, other feature properties, such as feature ranking scores, can be used to rank patches. For example, if Patch A with an average feature ranking score 4.6 and the best feature ranking score 4 of 14 associated features, and Patch B with an average feature ranking score 7.9 and the best feature ranking score 6 of 9 associated features, then Patch A is more important than Patch B for decision-making. To reliably rank patches, we defined different informative feature structures with useful feature properties as follows.

## 2.1 THE FEATURE SELECTION MAP

A FS method selects the top $k$ features from the $m$ features. The $k$ selected features have $k$ feature index numbers $I_p$ for $I_p \in \{0, 1, \ldots, m-1\}$ for $p = 0, 1, \ldots, k-1$. A top feature with $I_p$ is associated with a feature map $F^{q_p}$ where $q_p = I_p \bmod n$ for $p = 0, 1, \ldots, k-1$. Let $\bar{Q} = \{q_0, q_1, \ldots, q_{k-1}\}$. After eliminating duplicated elements in $\bar{Q}$, we get $Q$ with distinct elements for $Q \subseteq \bar{Q}$.

**Definition 1**: Let the feature selection map $T^l$ have features $t_{ij}^l$ for $i = 0, 1, \ldots, H-1$, $j = 0, 1, \ldots, W-1$, and $l = 0, 1, \ldots, n-1$. If $f_{ij}^l$ in a feature map $F^l$ is a selected feature, then $t_{ij}^l = f_{ij}^l$, otherwise $t_{ij}^l = 0$.

## 2.2 INFORMATIVE FEATURE MATRICES

Based on $n$ $H \times W$ feature maps $F^l$ for $l = 0, 1, \ldots, n-1$, five new definitions are given below for $i = 0, 1, \ldots, H-1$, and $j = 0, 1, \ldots, W-1$.

**Definition 2**: Let the "feature binary matrix" $B^l$ have binary numbers $b_{ij}^l$. If $f_{ij}^l$ is a selected feature, then $b_{ij}^l = 1$, otherwise $b_{ij}^l = 0$.

For a special case, the feature binary matrices $B^l$ with $b_{ij}^l = 1$ because all features in the $n$ feature maps are used.

**Definition 3**: Let the "feature accumulation matrix" $A$ have elements called "feature accumulators" $a_{ij}$, where $a_{ij} = \sum_{l=0}^{n-1} b_{ij}^l$.

**Definition 4**: Let the "feature distribution matrix" $D$ have elements $d_{ij}$ for, where $d_{ij} = a_{ij}/k$.

The features $t_{ij}^q$ of the feature selection map $T^q$ are ranked by a feature ranking method, such as the RFE (Guyon et al., 2002; RFE, 2024), then a feature $t_{ij}^q$ has its positive integer ranking number $\bar{r}_{ij}^q$ for $i = 0, 1, \ldots, H-1$, $j = 0, 1, \ldots, W-1$, and $q \in Q$, where the lower a ranking number, the higher a feature ranking. $\bar{r}_{ij}^q$ are sorted to generate new ranking numbers $r_{ij}^h$ in an increasing order for $h = 0, 1, \ldots, a_{ij}-1$.

**Definition 5**: Let the "feature ranking matrix" $R_k^h$ have $k$ positive integer ranking numbers $r_{ij}^h$ for top $k$ features with $k$ feature index numbers $I_p$ for $p = 0, 1, \ldots, k-1$ where $r_{ij}^h \leq r_{ij}^{h+1}$ for $i = \lfloor \frac{\mu}{W} \rfloor$, $j = \mu \bmod W$, and $h = 0, 1, \ldots, Max(a_{ij})-1$, where $\mu = \lfloor \frac{I_p}{n} \rfloor$. The smaller a positive integer ranking number, the higher the ranking of the top feature. Elements other than the $k$ positive integer ranking numbers of $R_k^h$ are 0 for $h = 0, 1, \ldots, Max(a_{ij})-1$.

**Definition 6**: Let the "average feature ranking matrix" $\bar{R}$ have average feature ranking values $\bar{r}_{ij}$ where $\bar{r}_{ij} = (\sum_{l=0}^{a_{ij}-1} r_{ij}^l)/a_{ij}$ for $i = 0, 1, and \ldots, H-1$, $j = 0, 1, \ldots, W-1$, where $a_{ij}$ are the feature accumulators of the feature accumulation matrix $A$.

## 2.3 INFORMATIVE HEATMAP MATRICES

A heatmap is represented by a heatmap matrix $V$ that has elements $v_{ij}$ for $i = 0, 1, \ldots, H-1$ and $j = 0, 1, \ldots, W-1$, where $0 \leq v_{ij} \leq 1$. A higher $v_{ij}$ makes a more impact on the decision. To get useful information associated with decisions of a CNN, we use a trained CNN to generate both feature maps and a decision that are used by a CAM-based method to generate $L$ heatmaps by using $L$ training data to extract activation strengths and activation frequencies of the elements of heatmap matrices $V_g$ for $g = 0, 1, \ldots, L-1$, and then rank the heatmap elements $v_{ij}$ for $i = 0, 1, \ldots, H-1$ and $j = 0, 1, \ldots, W-1$ in terms of importance associated with the decisions of a CNN. New definitions are given below.

**Definition 6**: Let the "heatmap count matrix" $C$ have elements $c_{ij}$ for $i = 0, 1, \ldots, H-1$ and $j = 0, 1, \ldots, W-1$, where $c_{ij}$ is the number of $v_{ij}^g$ where $v_{ij}^g > 0$ of $L$ heatmap matrices $V_g$ for $g = 0, 1, \ldots, L-1$.

**Definition 7**: Let the "heatmap activation matrix" $U$ have elements $u_{ij}$ for $i = 0, 1, \ldots, H-1$ and $j = 0, 1, \ldots, W-1$, $u_{ij} = c_{ij}/L$) where $c_{ij}$ is the number of $v_{ij}^g$ for $v_{ij}^g > 0$ of $L$ heatmap count matrices $V_g$ for $g = 0, 1, \ldots, L-1$.

**Definition 8**: Let the "heatmap strength matrix" $S$ have elements $s_{ij}$ for $i = 0, 1, \ldots, H-1$ and $j = 0, 1, \ldots, W-1$, where $s_{ij} = [\sum_{g=0}^{L-1} v_{ij}^g]/L$ for $L$ heatmap matrices $V_g$ for $g = 0, 1, \ldots, L-1$.

## 2.4 A New FS-CAM Method

Unlike traditional 2D CAM-based methods without FS (Zhou et al., 2016a; Selvaraju et al., 2017) and 3D CAM-based methods without FS such as 3DGradCAM (Williamson et al., 2022), we propose a new 2D FS-Grad-CAM method where we employ a FS method to select the top-ranked features from the flattened features first before applying FS-Grad-CAM for generating heatmaps. The traditional 2D methods without FS use equation (1) to calculate the neuron importance weights $w_l^c$ by using all $m$ features in $n$ $H \times W$ feature maps $F^l$ for $l = 0, 1, \ldots, n-1$.

$$w_l^c = \frac{1}{HW} \sum_{i=0}^{H-1} \sum_{j=0}^{W-1} \frac{\partial y^c}{\partial f_{ij}^l}, \tag{1}$$

where $y^c$ is the score for class $c$.

The new FS-Grad-CAM method uses a FS method to select the top $k$ features from $m$ flattened features. The new equation for calculating the neuron importance weights is shown in (2).

$$w_q^c = \frac{1}{HW} \sum_{i=0}^{H-1} \sum_{j=0}^{W-1} \frac{\partial y^c}{\partial t_{ij}^q}, \tag{2}$$

where $t_{ij}^q$ is an element of the feature selection map $T^q$ for $q \in Q$.

Final class scores are defined by

$$S^c = \sum_{q \in Q} \frac{w_q^c}{HW} \sum_{i=0}^{H-1} \sum_{j=0}^{W-1} t_{ij}^q \tag{3}$$

Finally, the saliency map $M_{FS-Grad-CAM}^c$ for an image is generated by equation (4).

$$M_{FS-Grad-CAM}^c = ReLU(S^c) \tag{4}$$

Since heatmap areas associated with fewer non-zero elements in a feature accumulation matrix with FS are more interpretable than those without FS, the FS-Grad-CAM using fewer features generates more explainable heatmaps with fewer areas than Grad-CAM using all features.

## 2.5 The 2D Patch Ranking Algorithm

The patches a $\bar{H} \times \bar{W}$ input image can be ranked based on degrees of importance for decisions. $\bar{H}$ is divisible by $H$ and $\bar{W}$ is divisible by $W$ for even patch distribution for the PRM.

**Definition 10**: The "patch ranking map" (PRM) is a $\bar{H} \times \bar{W}$ matrix having $P$ patches with patch ranking numbers $\lambda_{ij}$ for $P = HW$, and $\lambda_{ij} \in \{1, 2, \ldots, HW\}$ for $i = 0, 1, \ldots, H-1$, $j = 0, 1, \ldots, W-1$. The smaller $\lambda_{ij}$ is, the more important a patch at $(i, j)$ is associated with the decision.

A trained CNN generates $m$ flattened features from the $n$ $H \times W$ feature maps for $m = n \times H \times W$. The new training data with the $m$ flattened features are used for further FS. The five matrices (feature distribution matrix, feature ranking matrix $R_k^0$ for the top $k$ features, average feature ranking matrix, heatmap activation matrix, and heatmap strength matrix) related to decisions can be used to rank image patches to understand the relationship between image patches and final decisions. The top-ranked image patches are useful for a user, such as a medical doctor, to understand which image areas are most important for making a decision. A new 5-factor 2D image patch ranking algorithm is proposed by using 5 factors, as shown in Algorithm 1.

---

**Algorithm 1** The 2D patch ranking algorithm

---

**Input:** A feature distribution matrix $D$, a feature ranking matrix $R_k^0$ for the top $k$ features, the average feature ranking matrix $\bar{R}$, a heatmap activation matrix $U$, and a heatmap strength matrix $S$.

**Output:** A PRM.

1: Step 1: Calculate a ranking score $\theta_{ij}$ of a patch at $(i,j)$ where the monotonically non-decreasing function $\theta_{ij} = f(d_{ij}, r_{ij}^0, \bar{r}_{ij}, u_{ij}, s_{ij})$ for $i = 0, 1, \ldots, H-1$ and $j = 0, 1, \ldots, W-1$.

2: Step 2: Sort all patch ranking scores in a non-increasing order.

3: Step 3: Generate patch ranking numbers $\lambda_{ij}$ based on the non-increasing order.

4: Step 4: Generate a PRM using the $\lambda_{ij}$.

---

## 3 THE HYBRID BLOCK RANKING ALGORITHM

A $\bar{H} \times \bar{W} \times \bar{D}$ 3D input image has $P\,(\bar{H}/H) \times (\bar{W}/W) \times (\bar{D}/D)$ blocks for $P = HWD$ (assuming $\bar{H}$, $\bar{W}$, and $\bar{D}$ are divisible by $H$, $W$, and $D$, respectively). 3D image blocks can be ranked based on degrees of importance for decisions. A new definition is given next.

**Definition 11**: The "block ranking map" (BRM) is a $\bar{H} \times \bar{W} \times \bar{D}$ 3D input image having $P\,(\bar{H}/H) \times (\bar{W}/W) \times (\bar{D}/D)$ blocks with positive block ranking numbers $\phi_{ijk}$ for $P = HWD$ for $i = 0, 1, \ldots, H-1$, $j = 0, 1, \ldots, W-1$, and $k = 0, 1, \ldots, D-1$. The smaller $\phi_{ijk}$ is, the more important a block at $(i, j, k)$ is associated with the decision. The 3D image block ranking framework, as shown in Fig. 1, has 8 steps. These 8 steps using axial images are given as follows.

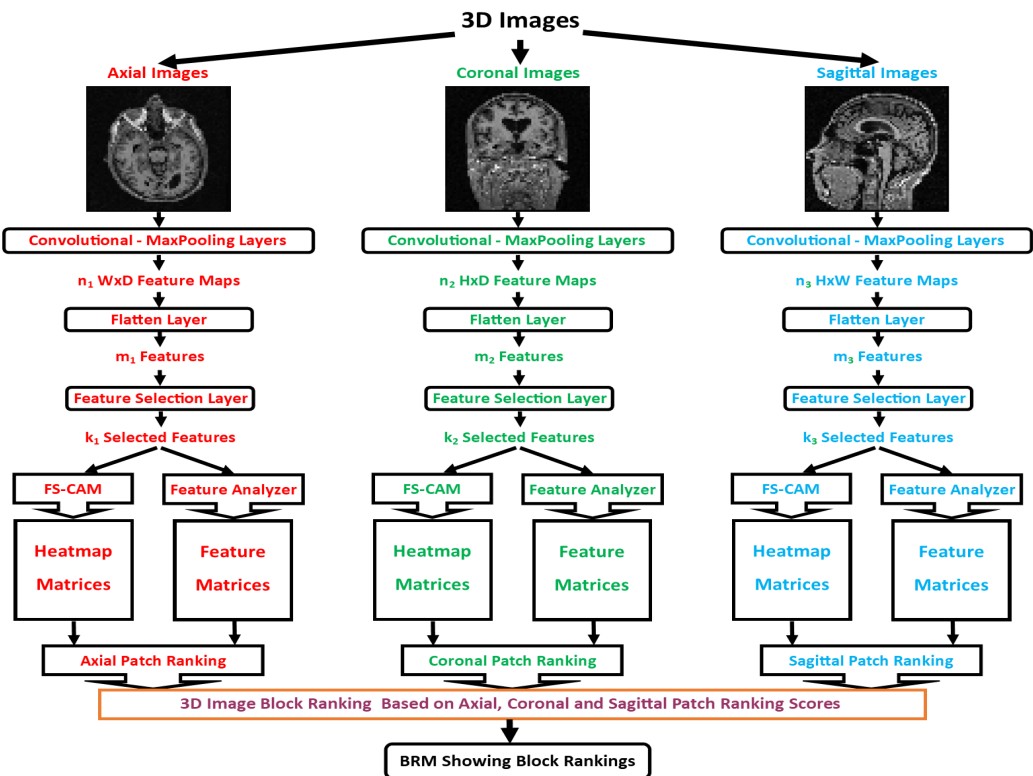

Figure 1: The 3D image block ranking framework for explainable medical imaging.

Step 1: axial images are extracted from the 3D images. Step 2: the 2D convolutional layers extract $n_1$ $W \times D$ axial feature maps from the axial images. Step 3: the flatten layer converts the axial feature maps to $m_1$ axial flattened features. Step 4: the FS layer selects the top $k_1$ axial features from the $m_1$ axial flattened features. Step 5: the FS-CAM method generates axial heatmap matrices. Step 6: the feature analyzer generates axial feature matrices. Step 7: the three 2D patch ranking

algorithms generate axial patch ranking scores, coronal patch ranking scores, and sagittal patch ranking scores, respectively. Step 8: the 3D image block ranking algorithm, as shown in Algorithm 2, generates different 3D image block ranking score matrices, and then the hybrid 3D image block ranking algorithm, as shown in Algorithm 3, finally uses them to generate a reliable hybrid BRM.

---

**Algorithm 2** The 3D image block ranking algorithm

---

**Input:** The axial patch ranking scores $\theta_{jk}$, coronal patch ranking scores $\theta_{ik}$, and sagittal patch ranking scores $\theta_{ij}$.

**Output:** A BRM.

 1: Step 1: If $\theta_{jk} > 0$, $\theta_{ik} > 0$, $\theta_{ij} > 0$, and the block $(i, j, k)$'s center is within a rational region such as a brain region, calculate a 3D image block ranking score $\varphi_{ijk}$ of a block at (i,j,k) where the monotonically non-decreasing function $\varphi_{ijk} = f(\theta_{jk}, \theta_{ik}, \theta_{ij})$ for $i = 0, 1, \ldots, H - 1$, $j = 0, 1, \ldots, W - 1$, and $k = 0, 1, \ldots, D - 1$.
 2: Step 2: Sort all patch ranking scores in a non-increasing order.
 3: Step 3: Generate block ranking numbers $\phi_{ijk}$ based on the non-increasing order.
 4: Step 4: Generate a BRM using the $\phi_{ijk}$.

---

"If $\theta_{jk} > 0$, $\theta_{ik} > 0$, $\theta_{ij} > 0$" in Step 1 of the 3D image block ranking algorithm is used to only select blocks with relevant axial, coronal and sagittal patches that are selected by the 2D patch ranking algorithm. In other words, other blocks with one relevant patch or two relevant patches are eliminated. Thus, Step 1 performs block selection. To reliably rank 3D image blocks to reduce the bias of one 3D image block ranking algorithm, the hybrid 3D image block ranking algorithm uses multiple 3D image block ranking algorithms using different top feature sets to generate different 3D image block ranking score matrices, and then generates a robust hybrid BRM.

---

**Algorithm 3** The hybrid 3D image block ranking algorithm

---

**Input:** Different 3D image block ranking score matrices generated by the 3D image block ranking algorithms under different conditions.

**Output:** A hybrid BRM.

 1: Step 1: Calculate average block ranking scores based on the different 3D image block ranking score matrices.
 2: Step 2: Sort all block ranking scores in a non-increasing order.
 3: Step 3: Generate block ranking numbers $\bar{\phi}_{ijk}$ based on the non-increasing order for $i = 0, 1, \ldots, H - 1$, $j = 0, 1, \ldots, W - 1$, and $k = 0, 1, \ldots, D - 1$.
 4: Step 4: Generate a hybrid BRM using the $\bar{\phi}_{ijk}$.

---

# 4 PERFORMANCE ANALYSIS USING 3D IMAGES FOR AD DIAGNOSIS

The ADNI (AD Neuroimaging Initiative) dataset with 982 3D brain images (ADNI, 2024; Amin, 2024) is used for three-class 3D image classification performance analysis. The 982 3D brain images include 284, 477, and 221 3D brain images for the cognitively normal (CN) class, mild cognitive impairment (MCI) class, and AD class, respectively. The 3D 982 brain images are resized to $64 \times 64 \times 64$ 3D brain images. A $64 \times 64 \times 64$ 3D brain image has $4,096$ $4 \times 4 \times 4$ blocks. $19,640$ axial images, $19,640$ coronal images, and $19,640$ sagittal images are extracted from the 982 3D images (i.e., 20 consecutive slices with indices from 22 to 41 are extracted from the middle of each 3D brain image). $13,748$ training images (i.e., 70% of the $19,640$ images) and $2,636$ testing images (i.e., 30% of the $19,640$ images) are used for simulations.

Three CNN models are trained by using $13,748$ $64 \times 64$ axial images, $13,748$ $64 \times 64$ coronal images, and $13,748$ $64 \times 64$ sagittal images, respectively. The three trained CNN models with testing accuracies 0.9309, 0.9367, and 0.9897 generate $64$ $16 \times 16$ axial feature maps, $64$ $16 \times 16$ coronal feature maps, and $64$ $16 \times 16$ sagittal feature maps, respectively. A $64$ $16 \times 16$ feature map has $16,384$ flatten features. Each element of the $16 \times 16$ feature accumulation matrix is 64. The $16 \times 16$ feature map has 256 features that are associated with 256 $4 \times 4$ patches in a $64 \times 64$ axial image, $64 \times 64$ coronal image, or $64 \times 64$ sagittal image. The $16,384$ flatten features with feature index numbers (i.e., 0, 1, ..., 16383) are used for further FS to eliminate less important features.

### 4.1 SELECTING RATIONAL TOP-RANKED FEATURES EFFICIENTLY

An axial image patch, a coronal image patch, and a sagittal image patch of a 3D image block with indices $(i, j, k)$ have indices $(j, k)$, $(i, k)$ and $(i, j)$ for $i = 0, 1, \ldots, 15$, $j = 0, 1, \ldots, 15$, and $k = 0, 1, \ldots, 15$, respectively. Axial, coronal or sagittal patches that are not associated with the standard brain of the "ebrains" software tool (i.e., their centers are not within the standard brain) are eliminated. A sample feature selection rule is given in Appendix A. $9,088$ axial features, $8,448$ coronal features, and $8,832$ sagittal features are eliminated from the $16,384$ axial features, $16,384$ coronal features, and $16,384$ sagittal features, respectively. Finally, $7,296$ rational axial features, $7,936$ rational coronal features, and $7,552$ rational sagittal features are added to the axial feature pool, coronal feature pool, and sagittal feature pool.

Three different FS methods using RFE and sklearn FS methods (Chi2, 2024; mutual_info_classif, 2024; f_regression, 2024; f_classif, 2024) are developed to generate three feature sets that are used to generate three feature matrices and three heatmap matrices from the three feature pools to finally get reliable patch rankings. The first FS method sequentially uses Chi2, mutual_info_classif, f_regression, f_classif, and RFE. The second FS method sequentially uses f_classif, mutual_info_classif, f_regression, Chi2, and RFE. The third FS method sequentially uses f_regression, mutual_info_classif, f_classif, Chi2, and the RFE.

Three axial 250-feature sets and three axial 100-feature sets are selected independently by using the three FS methods from the $16,384$ axial features. Three coronal 250-feature sets and three coronal 100-feature sets are selected independently by using the FS methods from the $16,384$ coronal features. Three sagittal 250-feature sets and three sagittal 100-feature sets are selected independently by using the FS methods from the $16,384$ sagittal features.

A $16 \times 16$ feature distribution matrix $D$, a $16 \times 16$ feature ranking matrix $R_k^0$ for the top $k$ features, the $16 \times 16$ average feature ranking matrix $\bar{R}$ for the top-ranked 250, and 100 features are generated. The $13,748$ training images are used to generate $13,748$ $16 \times 16$ heatmaps by using the FS-Grad-CAM. Then a heatmap activation matrix $U$ and a heatmap strength matrix $S$ are generated. Because of the rational FS method that uses index constraints for a sagittal image with indices $(i, j)$ to eliminate features out of the brain, all 250 top-ranked features are associated with the brain, as shown in a sagittal feature accumulation matrix (a number means how many top features are associated with the patch) and a sagittal heatmap (different colors are related to summations of a patch's positive values of all $13,748$ heatmaps) in Fig. 2(a) and Fig. 2(a), respectively.

All 256 elements of the $16 \times 16$ feature distribution matrix using all $16,384$ features shown in Fig. 2(c) have the same value 64 because all features in 64 $16 \times 16$ feature maps are used. The feature distribution matrix using all $16,384$ features cannot be used for ranking 256 patches. In addition, a heatmap using all $16,384$ features shown in Fig. 2(d) has many highlighted patches outside of the brain, such as colorful patches at two bottom corners and left-right sides; these patches cannot be used to rank patches. Thus, the feature distribution matrix and the heatmap using top 250 features in Fig. 2(a) and Fig. 2(b) are more rational and more interpretable for ranking patches than those using all $16,384$ features in Fig. 2(c) and Fig. 2(d). It is not reasonable to use all $16,384$ features, including features outside of the brain, to generate feature matrices. Thus, a patch ranking method using the three feature matrices and the two heatmap matrices should use the top-ranked features.

### 4.2 RELATIONSHIP BETWEEN TOP-RANKED BLOCKS AND RELEVANT BRAIN AREAS ASSOCIATED WITH AD DIAGNOSIS

The new 2D image patch ranking algorithm using the three feature matrices and the two 2D heatmap matrices to rank axial patches, coronal patches, and sagittal patches. The novel block ranking algorithm is used to generate the BRM by using the axial patch ranking scores, coronal patch ranking scores, and sagittal patch ranking scores together. Finally, the hybrid 3D image block ranking algorithm is used to generate 10 top-ranked blocks with block indices and world coordinates (in mm) of the standard brain in the "ebrains" software tool, as shown in Table 3 in Appendix C. Formulas for calculating world coordinates are given in Appendix B. Relevant brain areas are identified by using the "ebrains" software tool. To get reliable information for verifying if a brain area is associated with AD diagnosis, we used both ChatGPT (ChatGPT, 2024) and scientific literature (Mendonça et al., 2019; Traini et al., 2020). Brain areas related to the top 10 blocks are shown in Table 1. Table

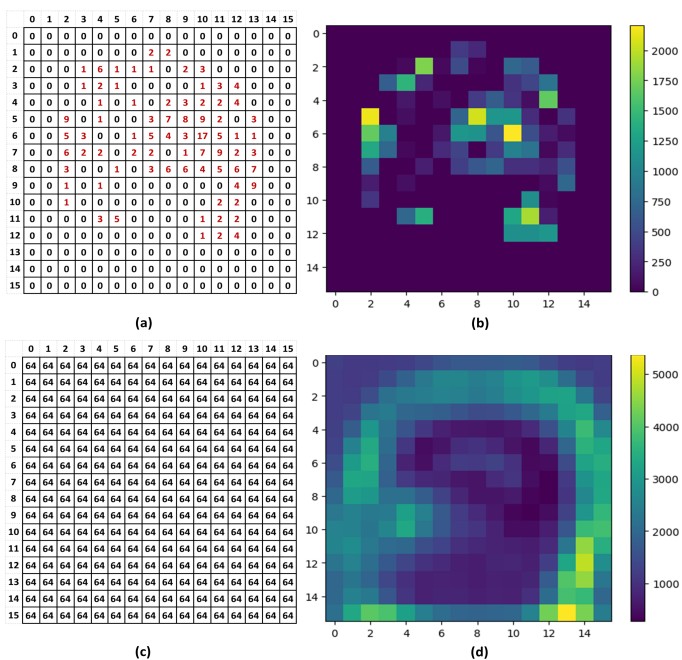

Figure 2: (a) A sagittal feature accumulation matrix with 250 top-ranked features, (b) a sagittal heatmap with 250 top-ranked features, (c) a sagittal feature accumulation matrix with all $16,384$ features, and (d) a sagittal heatmap with all $16,384$ features.

1 shows that 15 brain areas in black are associated with AD diagnosis, and one brain area in red is likely associated with AD diagnosis. For instance, ChatGPT states that CA1 (Hippocampus) left is indeed associated with AD diagnosis due to its susceptibility to the pathological changes that characterize the disease. Atrophy or structural changes in the left CA1 region of the hippocampus are associated with AD diagnosis. The CA1 subfield is highly vulnerable to neurodegenerative changes characteristic of AD.

Table 4 in Appendix D shows that all 10 brain areas related to the top 10 blocks are associated with AD diagnosis, and two brain areas are likely associated with AD diagnosis. Table 6 in Appendix E shows that 12 brain areas related to the top 10 blocks are associated with AD diagnosis, and two brain areas related to the top 10 blocks are likely associated with AD diagnosis. Therefore, the 3D image block ranking algorithm can discover important brain areas associated with AD diagnosis.

Table 1: The top 10 blocks and relevant brain areas (black: associated with AD diagnosis, red: likely associated with AD diagnosis).

| 1 | Frontal-to-Occipital right (Desikan et al., 2009; Pariente et al., 2005; Johnson et al., 1999) |
|---|---|
| 2 | Frontal-to-Occipital right |
| 3 | Frontal-to-Occipital left (Wang et al., 2023; Greicius et al., 2004; Zhou et al., 2024a) |
| 4 | hOc3d (Cuneus) right (Yang et al., 2019; Niskanen et al., 2011) |
| 5 | TE 3 (STG) left (Karas et al., 2007; Pariente et al., 2005), STS1 (STS) left (Thompson et al., 2001; Liebenthal et al., 2014), STS2 (STS) left (Thompson et al., 2001; Liebenthal et al., 2014) |
| 6 | TE 3 (STG) left, OP4 (POperc) left (Smith et al., 2018; Lee et al., 2020) |
| 7 | CA1 (Hippocampus) left (La Joie et al., 2013; Small et al., 2011; Kerchner et al., 2010), DG (Hippocampus) left (Bakker et al., 2012; Yassa et al., 2010; Kuhn et al., 2018), Frontal-to-Occipital left, FG3 (FusG) left (Karas et al., 2004; Dickerson et al., 2009; Shin et al., 2015), Ph1 (PhG) left (Pennanen et al., 2004; Karas et al., 2004; Frisoni et al., 2002) |
| 8 | hIP4 (IPS) right (Nelson et al., 2009; Li et al., 2012; Bai et al., 2009) |
| 9 | Temporal-to-Parietal right (Frisoni et al., 2010; Mosconi, 2005; Herholz et al., 2002) |
| 10 | TE 3 (STG) right (Karas et al., 2004; Teipel et al., 2007; Fan et al., 2011), STS1 (STS) right (Amlerova et al., 2022; Sacchi et al., 2023), Temporal-to-Parietal right |

Based on scientific literature (Braak & Braak, 1991; Gómez-Isla et al., 1996; Jack Jr. et al., 1997; Grady et al., 1988; Minoshima et al., 1997), and ChatGPT, the six most important brain areas associated with AD diagnosis include: Hippocampus, Entorhinal Cortex, Cerebral Cortex (Temporal, Parietal, and Frontal Lobes), Temporal Lobe, Parietal Lobe, and Frontal Lobe. Table 3 shown in Appendix B also shows the relationship between top 10 blocks and the six important brain areas including B1=Hippocampus, B2=Entorhinal Cortex, B3=Cerebral Cortex (Temporal, Parietal, and Frontal Lobes), B4=Temporal Lobe, B5=Parietal Lobe, and B6=Frontal Lobe.

Therefore, all 16 brain areas shown in Table 1 are associated with the six important brain areas associated with AD diagnosis. Thus, the hybrid 3D image block ranking algorithm can identify small 3D blocks in large 3D brain regions associated with AD diagnosis, such as the hippocampus.

### 4.3 VISUALIZING THE HYBRID BRM WITH AXIAL, CORONAL, AND SAGITTAL VIEWS

A medical doctor may conveniently use the hybrid BRM with axial, coronal, and sagittal 2D views to better understand the relationship between the top-ranked blocks and medical diagnosis so that the doctor can efficiently and effectively make a more explainable medical diagnosis.

For example, a patient's $256 \times 256 \times 256$ brain image has $4,096$ $16 \times 16 \times 16$ blocks. The top-ranked $16 \times 16 \times 16$ block at (6, 10, 9), as shown in Table 1, has 16 axial patches at (10, 9), 16 coronal patches at (6, 9), and 16 sagittal patches at (6, 10). A doctor can view the 16 axial patches, 16 coronal patches, and 16 sagittal patches for a rational diagnosis. For instance, the 9th axial, 9th coronal, and 9th sagittal patches are shown in Fig. 3. Other top-ranked blocks can be shown in axial, coronal, and sagittal 2D views for a doctor to analyze them and make a rational diagnosis.

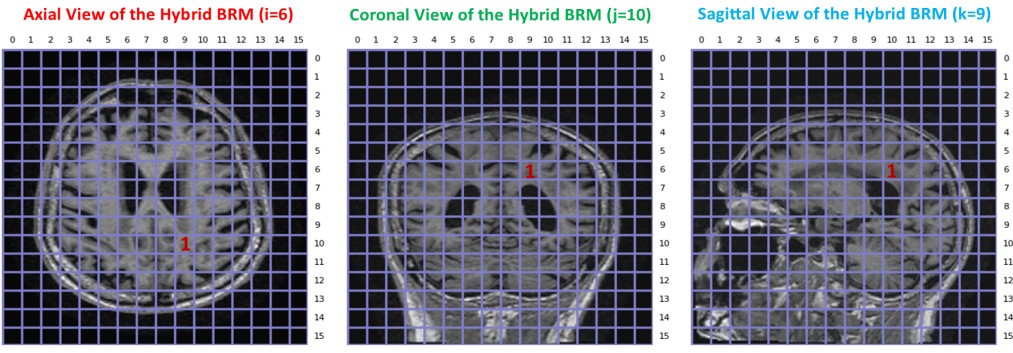

Figure 3: A Hybrid BRM with Axial, Coronal, and Sagittal Views for the Top Block at (6, 10, 9).

## 5 PERFORMANCE ANALYSIS USING 3D IMAGES FOR AUTISM DIAGNOSIS

286 3D brain images for autism diagnosis (binary classification) (Sujana, 2024) include 131 images for the autistic class and 155 images for the non-autistic class. They are resized to $64 \times 64 \times 64$ 3D images. $5,720$ axial images, $5,720$ coronal images, and $5,720$ sagittal images are extracted from the middle of 276 3D images brain images. $4,004$ training images and $1,716$ testing images are used for simulations. The FS method sequentially uses Chi2, mutual_info_classif, f_regression, f_classif, and RFE to select an axial 250-feature set, an axial 100-feature set, a coronal 250-feature set, a coronal 100-feature set, a sagittal 250-feature set, and a sagittal 100-feature set. These feature sets are then used to generate three feature matrices and two heatmap matrices. Then, the patch ranking algorithm generates the axial, coronal, and sagittal patch ranking scores. Finally, the hybrid 3D image block ranking algorithm is used to generate 10 top-ranked blocks, as shown in Table 2. Table 2 shows that 16 brain areas in black are associated with autism diagnosis, and one brain area in red is likely associated with autism diagnosis based on the cited publications and ChatGPT's answers. For instance, ChatGPT states that Frontal-I right is associated with autism spectrum disorder (ASD), and studies have observed reduced activation in the right IFG during tasks involving face processing in individuals with ASD. Thus, the hybrid 3D image block ranking algorithm is feasible and useful to identify important blocks associated with autism diagnosis.

Table 2: The top 10 blocks and relevant brain areas (black: associated with autism diagnosis, red: likely associated with autism diagnosis).

| 1 | Frontal-I right (Dapretto et al., 2006; Cai et al., 2014; Yang et al., 2015) |
|---|---|
| 2 | Ventral Dentate Nucleus (Cerebellum) left (Olivito et al., 2017; Arnold Anteraper et al., 2019; Jeong et al., 2012), Interposed Nucleus (Cerebellum) left (Zhou et al., 2024a;b; 2021), Dorsal Dentate Nucleus (Cerebellum) left (Olivito et al., 2017; Arnold Anteraper et al., 2019; Jeong et al., 2012) |
| 3 | Fo3 (OFC) right (Kendrick, 2023; Zikopoulos et al., 2020; Cheng et al., 2015), Temporal-to-Parietal right (Hu et al., 2021; Hao et al., 2022; Lombardo et al., 2011) |
| 4 | s24 (sACC) left (Simms et al., 2009; Zhou et al., 2016b; ETH Zurich, 2017) |
| 5 | Frontal-I right, 45 (IFG) right (Hadjikhani et al., 2007; Yang et al., 2015; Schmitz et al., 2014) |
| 6 | HC-Transsubiculum (Hippocampus) left (Dager et al., 2007; Bauman & Kemper, 2004; Utsunomiya et al., 2001), HC-Subiculum (Hippocampus) left (Dager et al., 2007; Bauman & Kemper, 2004; Utsunomiya et al., 2001), Ph3 (PhG) left (Mouga et al., 2022; Li et al., 2022; Postema et al., 2023), Frontal-to-Occipital left (Boets et al., 2018; Olive et al., 2022; Pugliese et al., 2019) |
| 7 | (FG1 (FusG) right, FG2 (FusG) right, FG3 (FusG) right, FG4 (FusG) right) Hadjikhani et al. (2004); Dalton et al. (2005); Nordahl et al. (2015), Ph2 (PhG) right (Zhang et al., 2023b; Mouga et al., 2022; McAlonan et al., 2009) |
| 8 | Temporal-to-Parietal right |
| 9 | Frontal-I right |
| 10 | Frontal-I right, 45 (IFG) right |

## 6 CONCLUSIONS

Both informative feature matrices and heatmap matrices generated by using top-ranked features are useful to reliably rank patches. The new FS-Grad-CAM method using top-ranked features, the new 2D image patch ranking algorithm using different top feature sets, and the novel 3D image block ranking algorithm using the axial, coronal, and sagittal patch ranking scores are able to generate relevant and useful information for robustly ranking 3D image block. The simulation results using the two different 3D data sets for AD diagnosis and autism diagnosis indicate that the novel hybrid 3D image block ranking algorithm can identify top-ranked blocks associated with important brain areas related to AD diagnosis and autism diagnosis. Thus, it is feasible and effective to robustly rank 3D image blocks by using the axial, coronal, and sagittal patch ranking scores together.

It is useful and efficient to use both ChatGPT and relevant publications together to reliably verify if a brain area is associated with a disease diagnosis. The hybrid BRM with axial, coronal, and sagittal 2D views of top-ranked 3D blocks is informative and convenient for a user to understand the relationship among 3D blocks, 2D patches, extracted feature maps, selected features, and final decisions. For example, a doctor may use the hybrid BRM with axial, coronal, and sagittal 2D views to conveniently, efficiently, and effectively make an explainable and rational medical diagnosis.

## 7 FUTURE WORKS

Firstly, we will develop more powerful FS methods to select top-ranked features that are used to generate highly informative feature matrices and heatmap matrices. Secondly, it is important to build an accurate deep learning model using the top-ranked features. Thirdly, the FS-Grad-CAM method, the 2D image patch ranking algorithm, and the hybrid 3D image block ranking algorithm will be improved by using other intelligent methods and optimized top feature sets. Fourthly, BRMs for correct decisions and BRMs for incorrect decisions will be generated to analyze the relationship among the top-ranked blocks, top-ranked features, relevant active elements in heatmaps, relevant brain areas, and final decisions. Fifthly, it is critical to find a precise function mapping indices $(i, j, k)$ of a 3D image block to corresponding world coordinates (I, J, K) of a 3D image block of the standard 3D brain of the "ebrains" software tool. Sixthly, a more effective 2D image patch ranking algorithm using more factors better than the 5-factor 2D image patch ranking algorithm will be developed. Seventhly, the hybrid 3D image block ranking algorithm will be evaluated by using other 3D data sets, such as lung cancer data. Finally, a new block ranking algorithm directly using 3D images, 3D deep learning with FS, and a 3D CAM-based method will be developed.

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

# A APPENDIX

For an axial image with indices $(j, k)$, the sample feature selection rule is If $((j = 2$ and $k \geq 5$ and $k \leq 10)$ or $(j = 3$ and $k \geq 4$ and $k \leq 11)$ or $(j = 4$ and $k \geq 3$ and $k \leq 12)$ or $(j = 5$ and $k \geq 3$ and $k \leq 12)$ or $(j = 6$ and $k \geq 3$ and $k \leq 12)$ or $(j = 7$ and $k \geq 2$ and $k \leq 12)$ or $(j = 8$ and $k \geq 2$ and $k \leq 13)$ or $(j = 9$ and $k \geq 2$ and $k \leq 13)$ or $(j = 10$ and $k \geq 2$ and $k \leq 12)$ or $(j = 11$ and $k \geq 3$ and $k \leq 12)$ or $(j = 12$ and $k \geq 4$ and $k \leq 11)$ or $(j = 13$ and $k \geq 5$ and $k \leq 10))$, Then the axial feature is added in a feature pool.

# B APPENDIX

Table 3: Relationship among top 10 blocks, their 3D indices $(i, j, k)$, and block centers' world coordinates (I, J, K) (in mm) of the standard brain, and the six most important brain regions.

| Block | $(i, j, k)$ | (I, J, K) | Score | B1 | B2 | B3 | B4 | B5 | B6 |
|---|---|---|---|---|---|---|---|---|---|
| 1 | (6, 10, 9) | (20.62, -53.76, 18.07) | 0.5918 | | | ✓ | | | ✓ |
| 2 | (5, 10, 9) | (31.04, -53.76, 20.62) | 0.5402 | | | ✓ | | | ✓ |
| 3 | (7, 10, 5) | (10.21, -53.76, -30.12) | 0.5394 | | | ✓ | | | ✓ |
| 4 | (7, 13, 10) | (10.21, -96.68, 30.12) | 0.5381 | | | ✓ | | | |
| 5 | (8, 8, 2) | (-0.21, -25.15, -66.26) | 0.5302 | | | ✓ | ✓ | | |
| 6 | (7, 7, 2) | (10.21, -10.85, -66.26) | 0.5241 | | | ✓ | ✓ | ✓ | |
| 7 | (8, 10, 5) | (-0.21, -53.76, -30.12) | 0.5227 | ✓ | ✓ | ✓ | ✓ | | ✓ |
| 8 | (7, 12, 10) | (10.21, -82.38, 30.12) | 0.5151 | | | ✓ | | ✓ | |
| 9 | (7, 11, 10) | (10.21, -68.07, 30.12) | 0.5143 | | | ✓ | ✓ | ✓ | |
| 10 | (8, 8, 13) | (-0.21, -25.15, 66.26) | 0.5076 | | | ✓ | ✓ | | |

# C APPENDIX

The 3D borders of the standard 3D brain and outside regions of the "ebrains" software tool are $Top$=88 mm, $Bottom$=-78 mm, $Front$=96 mm, $Back$=-132 mm, $Left$=-96 mm, and $Right$=96 mm. $Depth = Bottom–Top = -166$, $Length = Back–Front = -228$, and $Width = Right - Left = 192$. The center of a $4 \times 4 \times 4$ block with indices $(i, j, k)$ has voxel coordinates $i_{voxel\_center} = 4i + 2$, $j_{voxel\_center} = 4j + 2$, and $k_{voxel\_center} = 4k + 2$. The center of the $12mm \times 14.25mm \times 10.375mm$ block of the standard 3D brain of the "ebrains" software tool has corresponding world coordinates $i_{world\_center} = i_{voxel\_center} \times Width/63 + Left$, $j_{world\_center} = j_{voxel\_center} \times Length/63) + Front$, and $k_{world\_center} = k_{voxel\_center} \times Depth/63 + Top$.

# D APPENDIX

10 top-ranked blocks for 100 top features are generated by using three axial patch ranking scores, three coronal patch ranking scores, and three sagittal patch ranking scores. Table 3 shows relationship among the 10 top-ranked blocks, their 3D indices $(i, j, k)$, and block centers' world coordinates (I, J, K) used by the "ebrains" software tool. Brain areas related to top 10 blocks for 100 top features are all associated with AD diagnosis, as shown in Table 4.

# E APPENDIX

10 top-ranked blocks for the top 250 features are generated by using three axial patch ranking scores, three coronal patch ranking scores, and three sagittal patch ranking scores. Table 5 shows relationship among the 10 top-ranked blocks, their 3D indices $(i, j, k)$, and block centers' world coordinates (I, J, K) used by the "ebrains" software tool. Brain areas related to top 10 blocks for the top 250 features are all associated with AD diagnosis, as shown in Table 6.

Table 4: The top 10 blocks and relevant brain areas associated with AD diagnosis (100 top features).

| | | | |
|---|---|---|---|
| 1 | hOc3d (Cuneus) right | | |
| 2 | hIP4 (IPS) right | | |
| 3 | TE 3 (STG) left | STS1 (STS) left | STS2 (STS) left |
| 4 | Temporal-to-Parietal right | | |
| 5 | Frontal-to-Occipital left | | |
| 6 | Frontal-to-Occipital right | | |
| 7 | TE 3 (STG) right | STS1 (STS) right | STS2 (STS) right |
| 8 | Frontal-to-Occipital right | | |
| 9 | Temporal-to-Parietal right | Frontal-to-Temporal-II right | Frontal-to-Occipital right |
| 10 | Frontal-to-Occipital left | | |

Table 5: Relationship among 10 top-ranked blocks, their 3D indices $(i, j, k)$, and block centers' world coordinates (I, J, K) used by the "ebrains" software tool. (top 250 features)

| Block | $(i, j, k)$ | (I, J, K) | Score |
|---|---|---|---|
| 1 | (6, 10, 9) | (20.62, -53.76, 18.07) | 0.6803 |
| 2 | (6, 10, 6) | (20.62, -53.76, -18.07) | 0.6317 |
| 3 | (8, 10, 5) | (-0.21, -53.76, -30.12) | 0.5970 |
| 4 | (7, 10, 5) | (10.21, -53.76, -30.12) | 0.5862 |
| 5 | (5, 10, 9) | (31.04, -53.76, 18.07) | 0.5830 |
| 6 | (7, 7, 2) | (10.21, -10.85, -66.26) | 0.5822 |
| 7 | (6, 2, 9) | (20.62, 60.68, 18.07) | 0.5746 |
| 8 | (8, 7, 2) | (-0.21, -10.85, -66.26) | 0.5741 |
| 9 | (5, 7, 2) | (31.04, -10.85, -66.26) | 0.5552 |
| 10 | (5, 2, 9) | (31.04, 60.68, 18.07) | 0.5419 |

Table 6: The top 10 blocks and relevant brain areas (black: associated with AD diagnosis, red: likely associated with AD diagnosis). (Top 250 Features)

| | | | | | |
|---|---|---|---|---|---|
| 1 | Frontal-to-Occipital right | | | | |
| 2 | Frontal-to-Occipital left | | | | |
| 3 | CA1 (Hippocampus) left | DG (Hippocampus) left | Frontal-to-Occipital left | FG3 (FusG) left | Ph1 (PhG) left |
| 4 | Frontal-to-Occipital left | | | | |
| 5 | Frontal-to-Occipital right | | | | |
| 6 | TE 3 (STG) left | OP4 (POperc) left | | | |
| 7 | Frontal-I right | Fp2 (FPole) right | | | |
| 8 | TE 3 (STG) left | STS1 (STS) left | STS2 (STS) left | | |
| 9 | 1 (PostCG) left | OP4 (POperc) left | | | |
| 10 | p32 (pACC) right | Frontal-I right | | | |

