# OpenReview forum: "A New 3D Image Block Ranking Method Using Axial, Coronal and Sagittal Image Patch Rankings for Explainable Medical Imaging"
_ICLR.cc/2025/Conference — Submitted to ICLR 2025_

### Official Review · Reviewer_9Akt · 2024-10-24

**Soundness:** 1
**Presentation:** 1
**Contribution:** 2
**Rating:** 3
**Confidence:** 3

**Summary:**

This paper introduces a feature-selected (FS) Grad-CAM method to generate more focused explainable heatmaps with smaller highlighted areas. Additionally, a novel 2D image patch ranking algorithm was developed to reliably rank patches along the axial, sagittal, and coronal axes using features extracted by FS-Grad-CAM. These ranked scores are then used to create a Block Ranking Map (BRM) via a newly developed 3D block ranking algorithm. The resulting block-ranked scores are further refined through a novel hybrid 3D block ranking algorithm to produce a reliable hybrid BRM. The method was validated on Alzheimer’s Disease (AD) data and identified the top 10 ranked blocks associated with AD.

**Strengths:**

The paper used the 2D images in axial, sagittal and coronal axes to rank the 3D images, which makes the research novel.  This paper has detailed explanation on the algorithm, which makes the method replication easier.

**Weaknesses:**

1. Lack of competition: The cited papers (He et al., 2019 and Selvaraju et al., 2017) in the introduction part have compared their new algorithm (Grad-CAM) with other common machine learning algorithm (e.g. logistic regression) to demonstrate their better performance in the binary classification. But this paper doesn’t have compared with any baseline to demonstrate its superiority. There exists 3D medical imaging visual explanation algorithm (e.g. Respond-CAM) which might be a good benchmark to compare with.
2. Lack of generality: The paper only evaluated on one dataset, which cannot guarantee the generality of the proposed method. More tests are needed to justify the statement.

**Questions:**

1. It would be great if any qualitative/quantitative comparison with similar algorithm could be added to help the readers to better evaluate the performance of the proposed method.
2. It would be ideal if more dataset could be used to evaluate the performance of the proposed method (e.g. LUNA 16 lung nodule dataset)

---

> ### Author Response · Authors · 2024-11-17
>
> 1. It is an important suggestion! Since we have not  found block ranking methods in publications, we’ll continue to make efforts to find relevant methods, compare our method with them, and finally update the paper.
> 2. We truly agree. A new section (Section 5 “Performance Analysis Using 3D Images for Autism Diagnosis” on Page 9 was added in the revised paper. The additional simulation results indicate that the novel hybrid 3D image block ranking algorithm can also identify top-ranked blocks associated with important brain areas related to autism diagnosis.

---

### Official Review · Reviewer_vxh9 · 2024-10-31

**Soundness:** 3
**Presentation:** 2
**Contribution:** 2
**Rating:** 3
**Confidence:** 4

**Summary:**

Authors present a method for modifying Grad-CAM feature attribution maps that is able to identify the most important 'blocks' in a 3D MRI brain image. The method is applied to output features of a trained CNN by sectioning each 3D image axis into an equal number of patches. The features from Grad-CAM are passed through a feature selection (FS) method consisting of a combination of standard library functions including recursive feature elimination (RFE). The resulting `k` ranked features are accumulated axis- (or patch-) wise, and the patches from different combinations of FS methods are passed into final step that aggregates the 2D features into 3D block features. The blocks that contain the most patches with highest ranking features are selected as the most important blocks. Authors evaluate their method by verifying that the selected blocks correspond with those that are know to be important in the literature, as well as asking Chat-GPT. Author's provide clinical reasoning to explain the attributions.

**Strengths:**

* The target of the Authors' work is important and moves towards a more explainable and trustworthy result for use by clinicians.
* Authors are thorough in their definitions and attempt to give the reader the detail to reproduce their work.

**Weaknesses:**

* In general, this paper is badly formatted and overly verbose. The notation is difficult to keep consistent and often badly defined. Further comments to this are made in the `Questions` section below.
* Authors spend very little time reviewing prior work and setting their method in context - Authors should add explanations of related prior work including methods for using statistical analysis of Grad-CAM attribution and saliency maps to identify significant regions in input data.
* Authors inexplicably use Chat-GPT to test the reliability of their method by asking it to verify the important brain regions associated with AD - there is no explanation for this, and no reason to be doing that instead of actually asking clinicians.
* Authors present their method only on a single 3D MRI dataset and do not discuss its applicability to other modalities, or indeed any other domain. Authors cannot claim that this is a method for "Explainable Medical Imaging" when results on a single dataset are reported.

**Questions:**

### Major Comments
* The definitions and notation presented in the paper are cumbersome, and difficult to read. This starts with the statement `P (H-bar/H) x (W-bar/W) patches for P=HW` (line 92) that is repeated several times (187, 214 etc.) which could be replaced by a less obtuse definition of patch size. Authors also do not define _n_ in this Section 2. Authors should better define their variables. Additionally, the cumbersome notation in definitions 1-6 is largely unnecessary - Authors can simplify this section by removing superfluous 'definitions' and describing the process through which they yield the ranking matrices: this will avoid repetition of the `i` and `j` indices and shorten this bloated paper.
* On a similar note, Authors introduce additional notation for the 'top feature map' `T^Q`. This terminology is confusing. It is not a top "feature map", but rather an aggregated "top k features" map combining the Grad-CAM features selected by 'some feature selection method'. Authors should consider re-wording this.
* Lines 221-234 - Authors present their 8 steps for Image Block Ranking algorithm. This is presented badly - the reader is capable of understanding that the same steps are applied to the 3 different axis without making each step of this method so verbose. Figure 1 shows this much better in fact. Authors should describe steps 1-7 on a single axis to improve readability.
* Authors show a visualization of important blocks identified by their method in Figure 2b. The full Grad-CAM output without feature selection is shown in Figure 3b. It is evident that applying some thresholding to the full feature map, and even just applying the brain-boundary regional constraints, would yield a similar map to their own. Can author's comment on the significant differences between using the Grad-CAM values directly in this way, rather than the additional steps in their method? It would have been helpful to show some quantitative comparison in their results given this is supposed to be an extension of Grad-CAM -based methods.
* Why did Authors use Chat-GPT in their evaluations? What is the benefit of this over asking the clinicians which Authors repeated claim that this system is aimed at helping?
* Have Authors used this method on other domains or modalities to demonstrate its effectiveness?

### Minor Comments
* The paragraph from lines 57 contains a lot of repetition and should be pared down: this sentiment is reflected in many parts of the paper. The reader is clear that having better explainability is important and that a clinician can use this information to inform their diagnosis.
* Line 147 "... makes a more impact on the decision" - to what decision are the Authors referring?
* Line 148 "... we use a trained CNN to generate L heatmaps..." - the language here is confusing: Authors are referring to the model on which they are performing Grad-CAM, not a random trained model that generates heatmaps.
* Table 2 is unclear - why is much of the table blank?
* Line 489 - this is confusing: what is meant by the 'the 9th ... patches are shown'?. There are 16x16 patches in each of the 3 axis images shown, and a single patch at (6, 10, 9) is highlighted.
* Section 4.1 - it is absolutely unnecessary to bloat this section with the indicies that correspond to the brain boundary - put this in the appendix, or show it as an image if the Authors feel it adds to their explanation.

**Details Of Ethics Concerns:**

I question the validity of using Chat-GPT in this work - but no significant Ethics concerns.

---

> ### Author Response · Authors · 2024-11-17
>
> 1. Thanks! The revised paper had newly updated contents. Firstly, n was defined: “The last Maxpooling layer of a CNN generates n HxW feature maps F^l” . Secondly, definitions in Section 2 were simplified by removing repetition of the i and j indices and the l indices.
> 2. We replaced  'top feature map'  by the new feature selection map that had an updated definition in Section 2.1 on Page 3.
> 3. We revised steps 1-7 on pages 5-6 accordingly.
> 4. It is possible to apply some thresholding to the full feature map, and even just applying the brain-boundary regional constraints would yield a similar map. But each patch within the brain is still associated with 64 features (assuming 64 feature maps are generated), our method uses a feature selection (FS) method to select top-ranked features, the patches within the brain are associated with different features (0, 1, 2, …, 64 features). In this case, the heatmap using a small number of top-ranked features is better than the heatmap using all features.
> 5. To reliably find the associations between brain areas and a brain disease, using  both research results in relevant publications and information from ChatGPT is better than traditionally using the research results in relevant publications. Thus, we used both literature investigation and ChatGPT to verify if a brain area was associated with a disease diagnosis.
> 6. Yes. A new section (Section 5 “Performance Analysis Using 3D Images for Autism Diagnosis” on Page 9 was added in the revised paper. The additional simulation results indicate that the novel hybrid 3D image block ranking algorithm can also identify top-ranked blocks associated with important brain areas related to autism diagnosis.
> 7. We removed the repeated contents.
> 8. The decision is generated by the classifier using extracted features.
> 9. The paper has a revised sentence: “we use a trained CNN to generate both feature maps and a decision that are used by a CAM-based method to generate L heatmaps”.
> 10. Many blocks are associated with a small number of brain areas.
> 11. Since the block at (6, 10, 9) has 16 axial patches at (10, 9), 16 coronal patches at (6, 9), and 16 sagittal patches at (6, 10), the 9th axial patch, the 9th coronal patch, and the 9th sagittal patch are shown as three examples. A doctor may view 16 axial patches, 16 coronal patches, and 16 sagittal patches one by one.  The paper has a revised sentence: “the 9th axial, the 9th coronal, and the 9th sagittal patches are shown in Fig. 3”.
> 12. The content was moved to the appendix.

---

> > ### Comment · Reviewer_vxh9 · 2024-11-26
> > **Acknowledgement of Author Response**
> >
> > Thank you to the Authors for their timely response. This Reviewer appreciates the Authors' efforts to clarify their notation and improve the readability of their method. Authors can further simplify their algorithmic steps 1-7 described on Pages 5-6 by dropping the 'axial' qualifier, and explaining that these steps apply to axial, sagittal, and coronal views, the results of which are combined in the final steps.
> >
> > This Reviewer is still unconvinced that using Chat-GPT is at all relevant. If Authors require further validation of their top features beyond what is found in the literature, then they should ask clinicians to review their work rather than rely on the output of some language model.
> >
> > Thanks to the Authors for supplying further validation of their feature selection work by performing experiments aimed at detecting relevant brain regions associated with ASD diagnosis.
> >
> > The paper feels more accessible and clear after the Authors' changes, however, this Reviewer feels that the lack of comparison with other works makes it difficult to assess the true utility or novelty of this paper. The use of Chat-GPT in the evaluation is also concerning - this alone does not make this Reviewer confident enough in this work to change the previous rating.

---

> > > ### Author Response · Authors · 2024-12-01
> > >
> > > Thanks for the new comments and important suggestions.
> > >
> > > Firstly, we will continue to improve and simplify algorithmic steps 1-7 based on your comments.
> > > Secondly, we agree that clinicians can help to verify results. Actually, our hybrid method uses supporting information from both cited papers and ChatGPT to verify if a brain area is associated with AD diagnosis or autism diagnosis. In other words, using both cited papers and ChatGPT can generate more reliable results than only using cited papers.
> > > In addition,  using ChatGPT with intelligent information summarization is a really efficient and effective way for non-medical experts to quickly and easily verify the association between a brain area and a brain disease generated from cited papers.
> > >
> > > An example is given below.
> > > For the question: “Is CA1 (Hippocampus) left associated with AD diagnosis?”
> > > Answer 1 based on the cited paper (La Joie et al., 2013):
> > > “Highlights
> > > • Using 3 T MRI, hippocampal subfields were measured in aMCI, AD and SD and controls.
> > > • CA1 atrophy was found to be predominant in all patient groups.
> > > • CA1 volume was the best discriminating measure between controls and aMCI patients.
> > > • AD and SD differed in asymmetry and anterior-predominance, not in subfield atrophy.”
> > >
> > > Answer 2 based on the cited paper (Kerchner et al., 2010):
> > > “Memory loss is often the first and most prominent symptom of Alzheimer's disease (AD), coinciding with the spread of neurofibrillary pathology from the entorhinal cortex (ERC) to the hippocampus. The apical dendrites of hippocampal CA1 pyramidal neurons, in the stratum radiatum/stratum lacunosum-moleculare (SRLM), are among the earliest targets of this pathology, and atrophy of the CA1-SRLM is apparent in postmortem tissue from patients with mild AD.”
> > >
> > > ChatGPT’s answer:
> > > “Yes, the left CA1 region of the hippocampus is significantly associated with the diagnosis of Alzheimer's Disease (AD). The CA1 subfield is one of the earliest regions to exhibit neurodegeneration in AD. Atrophy and neuronal loss in this area correlate strongly with memory impairments, which are hallmark symptoms of the disease. Structural imaging of the left CA1 region can thus serve as a valuable biomarker for early diagnosis and progression monitoring of AD.”
> > >
> > > Based on both cited papers’ answers and ChatGPT’s answer, we have a conclusion: CA1 (Hippocampus) left associated with AD diagnosis as shown in Table 1. Similarly, other conclusions based on both cited papers’ answers and ChatGPT’s answers are shown in Table 1 and Table 2. Importantly, the conclusions based on supporting information confirmed by  both cited papers and ChatGPT are more reliable than those based on the information only from cited papers. Thus, results in Tables 1 and 2 are relatively reliable. Clinicians and experts will help to verify the results to generate more robust conclusions. Since clinicians and experts may not know the latest results, such as an association between a brain area and a disease due to a large number of publications, a hybrid method using information from multiple sources like cited papers, ChatGPT, and experts will be developed in order to obtain even more robust conclusions.
> > >
> > > Finally, for the weakness: “the lack of comparison with other works makes it difficult to assess the true utility or novelty of this paper”, we couldn’t find any 3D block ranking methods in publications. Also, since we didn’t have verification data sets such as 3-class 3D image classification data for training and testing, we had to find brain areas associated with a block, then verify if the brain areas are associated with AD or ASD, and finally evaluate our novel hybrid block ranking algorithm indirectly.
> > >
> > > In summary, to our best knowledge, it is the first work for ranking 3D blocks using the axial, coronal, and sagittal patch ranking scores. It has useful applications in explainable image classification like explainable brain imaging.

---

### Official Review · Reviewer_GAsu · 2024-11-03

**Soundness:** 3
**Presentation:** 3
**Contribution:** 2
**Rating:** 3
**Confidence:** 3

**Summary:**

In this paper, the authors propose a new method for explaninable medical imaging. By combining Grad-CAM-based feature selection, 2d image patch ranking, and 3d image block ranking together, the proposed method can effectively provide accurate medical diagnosis with explainability. They validate the proposed method on an ADNI dataset.

**Strengths:**

1. Overall, the whole framework makes sense and looks like a good solution in practice.

**Weaknesses:**

1. My major concern is that combining axial, coronal and sagittal images for 3d medical image analysis is not something new. Although the whole framework makes sense practically, there are no major innovations in my opinion.
2. Experimental section looks weak. There are no solid performance evaluation and comparison with SOTA methods.

**Questions:**

1. Is it possible to build 3d analysis directly whithout resorting to 2d analysis? Can we do convolution, feature selection, FS-CAM+feature analyzer, and block ranking directly on 3d images? What are the major advantages of do 2d image analysis first, then fuse them totether by 3d block ranking?

**Details Of Ethics Concerns:**

This study involves human data collection. So the authors are expected to address the privacy concerns in this submission.

---

> ### Author Response · Authors · 2024-11-17
>
> 1. Since we didn’t find published methods that use 2D image patch ranking algorithm to rank axial patches, coronal patches, and sagittal patches firstly, and then use axial patch ranking scores, coronal patch ranking scores, and sagittal patch ranking scores to rank 3D blocks indirectly, to our best knowledge, our 3D block ranking framework is new.
>
> 2. A new section (Section 5 “Performance Analysis Using 3D Images for Autism Diagnosis” on Page 9 was added in the revised paper. The simulation results using two different 3D data sets indicate that the novel hybrid 3D image block ranking algorithm can identify top-ranked blocks associated with important brain areas related to AD diagnosis and autism diagnosis.
>
> 3. It is a great question! Yes, it is possible to build a 3D block ranking model directly using 3D images.
>
> We plan to do the following 3 steps:
>
> Step 1: use the new 2D image patch ranking algorithm to rank axial patches, coronal patches, and sagittal patches, and then use axial patch ranking scores, coronal patch ranking scores, and sagittal patch ranking scores to rank 3D blocks indirectly.
>
> Step 2: develop a 3D block ranking algorithm to rank blocks directly using 3D images.
>
> Step 3: merge the two different block rankings to robustly generate hybrid block rankings.

---

### Official Review · Reviewer_SEPM · 2024-11-04

**Soundness:** 1
**Presentation:** 1
**Contribution:** 1
**Rating:** 5
**Confidence:** 4

**Summary:**

This paper attempt to provide a more explainable 3D GradCam map by combining ideas of GradCam in each of the respective projections (coronal, axial, sagittal) of MRI data with feature selection concepts. The feature maps produced in the maxpooling convolutional layers are used to derive various representations such as heatmap matrics, feature matrices, and the values within them are ranked individually per view and then combined into a 3D block ranking. The authors claim the resulting visualization gives better indication of the disease and demonstrate this on ADNI data for Alzheimer's disease.

The paper is poorly written and many of the details seem to be automatically written through a translation software or perhaps LLM judging by the language used. For example, reading the abstract had a lot of details  that typically seen in results sections later rather than focusing on a high-level summary of the approach.  Another example is a sentence in line 164-165 which reads "Different from traditional CAM-based methods without FS a new FS-Grad-CAM methods uses a FS method to select the top k features from m flattened features." - Is this referring to their proposed approach. Normally we would phrase it as "Unlike traditional CAM-based methods, we propose a new method called FS-Grad_CAM where we employ a feature selection method to select the top K feature from m flattened features first before applying GradCAM."

 Many details are unclear including the novelty with respect to other 3DGradCam methods (see eblow). Overall, it needs a major rewrite and using clinically relevant terminology with better motivation of the healthcare problem addressed.

**Strengths:**

Paper is about explainable AI showing clinicians relevant features useful for diagnosis in the 3D MRI images by taking slices in multiple views and offering top-ranked patches that are correlated with disease and understand the relationship between the top blocks and the decisions made by 3D CNN. As such, the paper attempts an important problem, namely, making the disease classification more explainable to clinicians. The familiar mechanisms of GradCam are used and attempt is made to process complex MRI datasets in multiple views. If the method could be clearly explained, one could even see value in the technique for 3DGradCam in general although other 3D GradCam tools are available. The main argument appears to be that GradCam should be applied after feature selection in the feature maps.

**Weaknesses:**

As mentioned above, the paper is poorly written to determine if the idea being proposed is a variation of 3D gradCam. No comparisons are made to any methods to even see its merit. Several open questions arise (see below).
There are several tools available for 3DGradCam such as these. If they are not relevant for comparison, it would be helpful to at least explain how your method differs from these.
https://github.com/fitushar/3D-Grad-CAM
https://www.researchgate.net/publication/357899396_Automated_grading_of_enlarged_perivascular_spaces_in_clinical_imaging_data_of_an_acute_stroke_cohort_using_an_interpretable_3D_deep_learning_framework

The paper in current problem needs a full rewrite starting with explanation of the MRI disease visualization problem, the role of existing 3D GradCam and the need for feature selection prior to GradCam. The whole idea of class activation maps was to allow us to see the rationale for the classification with the visualization itself in a way doing regional feature selection. By applying a separate feature selection operator a priori, what would be the impact on the gradient operators and the resulting activation maps?

ChatGPT is briefly mentioned and it is not clear what it is being used for.
What does it mean to say ChatGPT is used to verify if a brain area is associated with a disease? What is the prompt used? What are the input, only text or text and image, is a bounding box and a prompt given as input. How accurate is ChatGPT in identifying  the brain areas associated with the disease. All these should be added to explain the use of ChatGPT.

**Questions:**

1. What is the rationale for a top-ranked 3Dimage block being correlated with diseases? Real-life experiences with 2d heat maps alone indicate they are not always a reliable indicator of a disease. Since a disease may be seen in some view better than other views, the method of fusion is important between the 3 views. Provide more discussion or justification for the correlation between top-ranked blocks and diseases, and the fusion method would help clarify better.

2. How is the ranking of the blocks done? In general, what does ranking mean in your context, is it just a matter of selecting high-valued entries in the feature and heat map matrices?

3. What is the purpose of ChatGPT in the work? It is said it is to verify if a brain area is associated with a disease. How does it work? Are both prompt text and image supplied as input? How accurate is ChatGPT in identifying  the brain areas associated with the disease. This discussion is brief and unconvincing so elaborating on how exactly ChatGPT is used and what its inputs are would explain this section better.

---

> ### Author Response · Authors · 2024-11-17
>
> 1. The revised abstract has no simulation details. The revised paper has a new sentence: “Unlike traditional 2D CAM-based methods without FS (Zhou et al., 2016a; Selvaraju et al., 2017) and 3D CAM-based methods without FS such as 3DGradCAM (Williamson et al., 2022), we propose a new 2D FS-Grad-CAM method where we employ a FS method to select the top-ranked features from the flattened features first before applying FS-Grad-CAM for generating heatmaps.”
>
> 2. In Section 2 on Page 2 in the revised paper, three new paragraphs after the first paragraph were added to explain (1) the relationship among input blocks, feature maps, selected features and decisions, and (2) the rationale for defining the new feature matrices. Section 2.4 on Page 4 states that 3D CAM-based methods without FS such as 3DGradCAM used in a cited paper are different from the new 2D FS-Grad-CAM method with FS.  The cited paper is “Automated grading of enlarged perivascular spaces in clinical imaging data of an acute stroke cohort using an interpretable, 3D deep learning framework”. In summary, a feature selection algorithm selects a small number of top-ranked features top-ranked features that are used to generate the three feature matrices and the two heatmap matrices that are used for ranking patches directly and ranking blocks indirectly, the top-ranked features are associated with a small number of relevant 3D blocks, since the top-ranked features used by a classifier are related to a disease diagnosis, so the relevant 3D blocks are also related to the disease diagnosis.
>
> 3. Since a block is associated with axial patches, coronal patches and sagittal patches, the block can be ranked indirectly by using them. Since a high-ranking block is associated with the high-ranking axial patch, the high-ranking coronal patch, and the high-ranking sagittal patch, the new 2D image patch ranking algorithm is developed to rank axial patches, coronal patches, and sagittal patches based on both selected features and heatmaps generated by using the selected features, and then axial patch ranking scores, coronal patch ranking scores, and sagittal patch ranking scores are used to rank 3D blocks.
>
> 4. To reliably find the associations between brain areas and a brain disease, using  both research results in relevant publications and information from ChatGPT is better than traditionally using the research results in relevant publications. Thus, we used both literature investigation and ChatGPT to verify if a brain area was associated with AD diagnosis and autism diagnosis that was discussed in a newly added section (Section 5 “Performance Analysis Using 3D Images for Autism Diagnosis” on Page 9) .

---

> > ### Comment · Reviewer_SEPM · 2024-11-29
> > **Response to rebuttal**
> >
> > I am still not convinced that the ChatGPT way is the best way to utilize information about the brain sturctures. A simple lookup table may have sufficed for this purpose unless the number of brain regions being considered is very large (which I am skeptical is the case).
> > I have upgraded the score to reflect the author's clarifications.

---

> > > ### Author Response · Authors · 2024-12-01
> > >
> > > Thanks for more suggestions.
> > >
> > > We agree with you: only using ChatGPT is not the best way. Thus, our hybrid method is not solely relying on ChatGPT. Actually, our hybrid method used supporting information from both cited papers and ChatGPT to verify if a brain area was associated with AD diagnosis or autism diagnosis. Using both cited papers and ChatGPT can generate more reliable results than only using cited papers. In addition,  using ChatGPT with intelligent information summarization is a really efficient and effective way for non-medical experts to quickly and easily verify the association between a brain area and a brain disease generated from cited papers.
> > >
> > > An example is given below.
> > > For the question: “Is CA1 (Hippocampus) left associated with AD diagnosis?”
> > > Answer 1 based on the cited paper (La Joie et al., 2013):
> > > “Highlights
> > > • Using 3 T MRI, hippocampal subfields were measured in aMCI, AD and SD and controls.
> > > • CA1 atrophy was found to be predominant in all patient groups.
> > > • CA1 volume was the best discriminating measure between controls and aMCI patients.
> > > • AD and SD differed in asymmetry and anterior-predominance, not in subfield atrophy.”
> > >
> > > Answer 2 based on the cited paper (Kerchner et al., 2010):
> > > “Memory loss is often the first and most prominent symptom of Alzheimer's disease (AD), coinciding with the spread of neurofibrillary pathology from the entorhinal cortex (ERC) to the hippocampus. The apical dendrites of hippocampal CA1 pyramidal neurons, in the stratum radiatum/stratum lacunosum-moleculare (SRLM), are among the earliest targets of this pathology, and atrophy of the CA1-SRLM is apparent in postmortem tissue from patients with mild AD.”
> > >
> > > ChatGPT’s answer:
> > > “Yes, the left CA1 region of the hippocampus is significantly associated with the diagnosis of Alzheimer's Disease (AD). The CA1 subfield is one of the earliest regions to exhibit neurodegeneration in AD. Atrophy and neuronal loss in this area correlate strongly with memory impairments, which are hallmark symptoms of the disease. Structural imaging of the left CA1 region can thus serve as a valuable biomarker for early diagnosis and progression monitoring of AD.”
> > >
> > > Based on both cited papers’ answers and ChatGPT’s answer, we have a conclusion: CA1 (Hippocampus) left associated with AD diagnosis as shown in Table 1. Similarly, other conclusions based on both cited papers’ answers and ChatGPT’s answers are shown in Table 1 and Table 2. Importantly, the conclusions based on supporting information confirmed by both cited papers and ChatGPT are more reliable than those based on the information only from cited papers. Thus, results in Tables 1 and 2 are relatively reliable. Clinicians and experts will help to verify the results to generate more robust conclusions. Since clinicians and experts may not know the latest results, such as an association between a brain area and a disease due to a large number of publications, a hybrid method using information from multiple sources like cited papers, ChatGPT, and experts will be developed in order to obtain even more robust conclusions.

---

### Official Review · Reviewer_yQWZ · 2024-11-05

**Soundness:** 2
**Presentation:** 1
**Contribution:** 2
**Rating:** 3
**Confidence:** 2

**Summary:**

This paper aims to tackle the limitation of explainability for medical image analysis, the contribution can be summarized as follows:
1) Proposed a new Grad-CAM-based method using feature selection to produce explainable heatmaps with a small number of highlighted image patches corresponding to top-ranked features
2) Designed a new 2D image patch ranking algorithm

**Strengths:**

The strength of this paper can be summarized as follows:
1) The problem that this paper aims to tackle is important
2) The adaptation of feature selection with Grad-CAM seems to be an interesting method

**Weaknesses:**

The weakness of this paper can be summarized as follows:
1) The writing and organization is poor
2) The innovation cannot be easily understand and a lot of definition with unclear steps are demonstrated in the first two sections
3) Clarity on the innovation is not clear

**Questions:**

1) Can you clarify your innovation with a simple paragraphs? A lot of definitions are demonstrated from section 2.1 to 2.3, in which they are not interconnected to demonstrate a clear step-by-step story towards section 2.4.
2) For 3D image block ranking, I am wondering can you directly use sliding window to extract patches for ranking? Seems like you are ensembling all different 2D orientated slices result. It will be great to clarify why you want to do it in this way. Is there any advantage on this?

---

> ### Author Response · Authors · 2024-11-17
>
> 1. The innovations were given in Abstract and Section 6 Conclusions on Page 10 in the revised paper. They include (1) developing a new 2D image patch ranking algorithm using both the three newly defined feature matrices that are defined in Sections 2.1 to 2.3 and the two newly defined heatmap matrices related to Section 2.4 to reliably rank axial patches, coronal patches, and sagittal patches, respectively, and (2) creating a novel 3D image block ranking algorithm to generate a ``Block Ranking Map (BRM)" by using the axial patch ranking scores, coronal patch ranking scores, and sagittal patch ranking scores generated by the new 2D image patch ranking algorithm. In Section 2 on Page 2 in the revised paper, three new paragraphs after the first paragraph were added to explain the rationale for defining the new feature matrices.
>
> 2. Yes. Actually, 20 consecutive axial slices, 20 consecutive coronal slices and 20 consecutive sagittal slices with indices from 22 to 41 are extracted from the middle of each 64x64x64 3D brain image for ranking patches. Section 7 Future Works on Page 10 added a new future work (i.e. “a new block ranking algorithm directly using 3D images, 3D deep learning with FS and a 3D CAM-based method will be developed”) . The 3D block rankings based on the 2D information will be compared with the 3D block rankings based on the 3D information. The two different block rankings will be merged to generate reliable final block rankings.

---

### Meta-Review · Area_Chair_vuXp · 2024-12-17

**Metareview:**

The paper aims to improve explainability in medical image analysis by proposing a Grad-CAM-based method enhanced with feature selection to produce focused heatmaps. This approach intends to make disease classification for 3D MRI images more interpretable for clinicians, which is a relevant and practical contribution. Reviewers acknowledged the importance of explainability in medical diagnostics and found the adaptation of feature selection with Grad-CAM to be an interesting direction.

However, the paper faces significant limitations. Reviewers unanimously raised concerns about the clarity of the writing, which made it difficult to identify the novelty of the method. Additionally, the evaluation is limited to a single 3D MRI dataset, weakening the experimental evidence and generalisability of the proposed approach. Given the combination of unclear novelty, limited experimentation, and reviewer consensus, I recommend rejection.

**Additional Comments On Reviewer Discussion:**

Reviewers unanimously recommended rejection after reading the rebuttal.

---

### Decision · Program_Chairs · 2025-01-22

Reject